# A Biopsychosocial Model of Sex Differences in Children’s Eating Behaviors

**DOI:** 10.3390/nu11030682

**Published:** 2019-03-22

**Authors:** Kathleen L. Keller, Samantha M. R. Kling, Bari Fuchs, Alaina L. Pearce, Nicole A. Reigh, Travis Masterson, Kara Hickok

**Affiliations:** 1Department of Nutritional Science, The Pennsylvania State University, University Park, PA 16802, USA; smk398@psu.edu (S.M.R.K.); baf44@psu.edu (B.F.); azp271@psu.edu (A.L.P.); nar5235@psu.edu (N.A.R.); kih101@psu.edu (K.H.); 2Department of Food Science, The Pennsylvania State University, University Park, PA 16803, USA; 3Department of Epidemiology, Geisel School of Medicine at Dartmouth College, Hanover, NH 03756, USA; travis.d.masterson@dartmouth.edu

**Keywords:** sex differences, eating behavior, food intake, biopsychosocial, children, brain imaging

## Abstract

The prevalence of obesity and eating disorders varies by sex, but the extent to which sex influences eating behaviors, especially in childhood, has received less attention. The purpose of this paper is to critically discuss the literature on sex differences in eating behavior in children and present new findings supporting the role of sex in child appetitive traits and neural responses to food cues. In children, the literature shows sex differences in food acceptance, food intake, appetitive traits, eating-related compensation, and eating speed. New analyses demonstrate that sex interacts with child weight status to differentially influence appetitive traits. Further, results from neuroimaging suggest that obesity in female children is positively related to neural reactivity to higher-energy-dense food cues in regions involved with contextual processing and object recognition, while the opposite was found in males. In addition to differences in how the brain processes information about food, other factors that may contribute to sex differences include parental feeding practices, societal emphasis on dieting, and peer influences. Future studies are needed to confirm these findings, as they may have implications for the development of effective intervention programs to improve dietary behaviors and prevent obesity.

## 1. Introduction

Sex and gender are important characteristics that contribute to individual variability in the development of disordered eating and obesity, but the extent to which they impact eating behaviors in children is less clear. It has been assumed that sex differences in eating behavior arise in adolescence because of the physiological changes and sociocultural pressures experienced during this developmental period. Prior to adolescence, sex-based influences on eating behavior have been thought to be minimal. However, there are both biological (e.g., sexual dimorphic patterns of in utero neural development and genetics) and psychosocial (e.g., parental feeding practices and societal body ideals) factors that may affect the way children eat prior to puberty. Despite these potential influences, this period of development has received little attention in the literature. Because of the sex differences that occur in the prevalence of both disordered eating [1,2,3] and obesity [4,5], there is a need to understand the role of sex in the development of behaviors involved with the etiology of these diseases prior to puberty. To call attention to this gap, this paper reviews the extant literature and presents new data demonstrating that sex differences in eating behavior arise prior to puberty and have effects on children’s appetitive traits and neural responses to food cues. 

The National Academy of Sciences has outlined rationale for when sex differences should be studied [6]. Several of their criteria apply to eating and weight disorders and therefore are relevant to the current paper. The first criterion is if there are known sex differences in the prevalence or incidence of a disease. Eating disorders occur nearly eight times more frequently in females than males [1,2,3]. At the same time, data from the National Health and Nutrition Examination Survey (NHANES) show across all age groups a higher prevalence of obesity among male children compared to females [7]. These striking statistics provide support for studying the role of sex in eating behaviors because they are integral to the development of these conditions. Another criterion outlined in this report is if there are known sex differences in disease severity, progression, or outcome. In the case of obesity, there are well-described differences in body composition, with adult males carrying fat around the abdomen and chest (i.e., visceral adipose tissue), which is associated with higher metabolic risk, while some pre-menopausal adult females are metabolically protected by accumulating fat in the lower extremities [8,9]. In addition, males tend to have more fat free mass than females. These differences are present in infancy [10,11,12] and persist throughout development, becoming more robust at puberty [13]. Furthermore, symptomology associated with binge eating (i.e., frequency and level of distress) is more severe in females relative to males [14]. A final criterion suggested in the report is if sex influences the success or outcome of interventions [6]. In both children [15] and adults [16], males tend to be more responsive to weight loss interventions than females. With the potential promise of personalized medicine for treatment of complex diseases such as obesity, understanding how sex influences response to treatment could highlight novel therapies that could specifically be targeted to males or females. 

Before reviewing the literature, it is worth noting that much of the research in this area has not distinguished between the constructs of “sex” and “gender”. Sex refers to the biological classification of male or female according to chromosomes and reproductive organs. Gender, on the other hand, refers to one’s self-representation, which is influenced by sociological and cultural factors [17]. Often one’s biological sex matches with self-assigned gender, but this is not always the case. The multitude of factors influencing both sex and gender have made the study of individual differences between males and females complicated. Because we are applying a biopsychosocial framework to describing how sex influences eating behavior, we include discussion of biological factors more likely to influence sex and social and psychological factors more likely to influence gender. However, as most prior studies do not clarify whether they distinguished between the two constructs when collecting participant data, it is not possible to make clear distinctions about how the terminology is used when referring to these studies. To avoid switching between “sex” and “gender” throughout the paper, we use the term “sex” as a combined term that includes not only biological, but also social and psychological influences. 

The goal of this paper is to present evidence that sex influences eating behaviors in childhood and to examine the source of these influences using a biopsychosocial framework. In the last section of the paper, we present new data analyses that have been informed by the biopsychosocial model. This paper is not intended to be a systematic review, but rather is a starting point for framing research questions that can systematically address the role of sex in childhood eating behavior. To support the argument that sex differences in childhood eating behavior are relevant to the development of eating and weight-disorders, we have selectively focused on some aspects of eating behavior (i.e., food acceptance, food intake, picky eating, appetitive traits, eating compensation, eating in the absence of hunger, and meal-specific microstructural patterns (e.g., bite rate and eating speed)). However, other important contributors to eating behavior that may not as directly impact risk for eating and weight disorders (e.g., oral sensory responses and olfactory sensations) have been omitted. Additionally, to avoid the inclusion of effects on eating behavior that could be influenced by the physiological and hormonal events related to puberty, to the extent possible, the literature review focuses on children age 11 years and younger, although it is recognized that this may not fully eliminate pubertal influences. However, due to the paucity of evidence in some sections, we have included a few studies that report on an age range beyond 11 years, although we recognize that the results may be influenced by pubertal development. Within the age group of children discussed, infants are defined as <1 year, toddlers as 1–2 years, preschool children as 3–5 years, and middle childhood as 6–11 years.

## 2. Evidence for Sex Differences in Children’s Eating Behavior 

The first part of the paper provides an overview of the available literature to support the role of sex in childhood eating behaviors. This section is divided into studies that have examined sex differences in food acceptance, food intake, appetitive traits, eating-related compensation, eating in the absence of hunger, and meal-specific microstructure. 

### 2.1. Sex Differences in Food Acceptance/Preference

The literature has consistently shown sex differences in children’s food acceptance and preference patterns, particularly for foods that impact weight status and overall dietary quality (i.e., fruits, vegetables, proteins, etc.). For food acceptance patterns, Cooke and colleagues [18] found that females (ages 4–7 years) liked a greater number of foods than male children. With regards to specific foods or food groups, studies including children from various countries have shown that females rate liking of fruits [18,19,20,21] and vegetables [18,19,20,21,22,23,24] higher than males, while male children report higher liking for meat, fish, poultry, and high-fat foods compared to females [18,19,20,25]. Furthermore, male children in middle childhood have higher acceptance of fatty and sugary foods [18] and foods and beverages characterized as “unhealthy” (e.g., sweet snacks, savory snacks, and sugar sweetened beverages) compared to female children [20]. Additionally, females in middle childhood show increased liking for vegetables [22] while males have greater liking for meat products [18]. 

While the aforementioned studies demonstrate sex differences in food acceptance in middle childhood, studies in toddlers and preschool-aged children have shown no differences [26,27]. However, it is not clear if null findings are in part due to a lack of sensitivity in the methods available to measure liking in preschool children (i.e., hedonic facial scales and parental report). These results demonstrate that. in middle childhood, females typically like or prefer foods that are often regarded as lower in energy and nutrient dense, such as fruits and vegetables, whereas males tend to like meats, meat products, and foods high in fat and sugar. The sensory and/or nutritional characteristics of the foods that drive these sex-effects are not known.

### 2.2. Sex Differences in Dietary Patterns

As liking and preference are primary determinants of what children eat [28,29], it is likely that sex also influences children’s dietary intake. This is especially apparent for fruits and vegetables [28]. In children as young as two years, intake of vegetables [30,31,32,33], fruits [24,31,32,34] and fruits and vegetables combined [31,35,36,37] is higher among females than males. Female children have similarly reported greater intake of foods classified as “healthy” and lower intake of “unhealthy” foods when compared to males [32]. Since these studies used self- and parentally-reported measures of food intake, there is potential for response bias as fruit and vegetable intake is a socially desirable behavior. However, studies using more objective assessment methods in schools have also observed that female students are more likely to consume from a salad bar than males [38,39]. The alignment with observational data strengthens the findings from questionnaires, suggesting that female children tend to consume more fruits and vegetables than males. 

In addition to fruits and vegetables, self-reported intake of other foods and food groups also varies by sex. In cohorts of European children, males report consuming more sugar and sweets [36,40], breakfast cereals, full-fat milk, meats/meat products, and baked beans while females consumed more oily fish, eggs, and cheese [36]. In the United States, male children tend to have higher intake of most food groups, as well as higher overall energy intake [37,41], although overall variety of foods consumed tends to be higher in females [42]. This finding supports the previously discussed observations that found females also *liked* a greater number of foods than male children [18]. Although these studies provide support for the notion that sex differences in eating behavior arise in childhood, not all studies agree [27,43]. Inconsistencies across studies could be due to variability in how dietary intake is measured (e.g., 24-h recall, food frequency, and direct observation), who is reporting dietary intake (e.g., parent vs. child), and the age and cognitive abilities of the child being studied [44]. There is a need to conduct more observational studies where food intake is directly measured to confirm sex-effects on reported intake in children. 

### 2.3. Sex Differences in Questionnaire Measures of Appetitive Traits

The literature reviewed in the preceding sections on both food acceptance and intake supports the notion that female children like and consume more foods that are typically thought to be protective against excess weight, e.g. fruits and vegetables. However, these associations between sex and liking/preference do not provide insight into why females are at greater risk for eating and weight disorders. An additional possibility is that females differ from males in appetitive traits that might make them more susceptible to eating in response to external food cues or less susceptible to feedback from internal satiety cues. In the following section, evidence for sex differences in parentally reported measures of eating behaviors and appetitive traits are presented. 

#### 2.3.1. Picky Eating

Picky eating is commonly observed in young children [45] and is associated with lower consumption of fruits and vegetables [30,46]. In the literature, picky eating has been conceptualized by two related constructs: (1) food neophobia, which is the rejection of novel or unknown foods; and (2) food fussiness, which is the rejection of many known, familiar foods. Whether the prevalence of picky eating differs by sex is unclear. No sex differences were evident for picky eating, more generally, in a sample of Canadian preschool-aged children [47] or in a review of studies in toddlers (≤30 months) [45]. For food neophobia, a study in French toddlers found males to have higher neophobia than females [48], however, other studies did not find sex differences [49,50]. In contrast, food fussiness, an eating trait assessed with the Children’s Eating Behavior Questionnaire (CEBQ) [51], has shown more consistent sex differences, however, the pattern of results is inconsistent. Males have been reported as fussier eaters than females in a cohort of 2–7-year-olds from the United Kingdom [23] and in 6–7-year-olds from the Netherlands [52], while, in toddlers from China, females were reported to be fussier eaters [53]. In general, there appears to be greater evidence for picky eating in males than females, but the inconsistent findings emphasize the need to delineate the underlying constructs, examine potential confounding factors (e.g., parental characteristics, and child age and temperament), and have appropriately powered samples (i.e., not over or under powered). 

#### 2.3.2. Appetitive Traits

Other studies have investigated whether there are sex differences among other appetitive traits assessed by the CEBQ [51]. Using this instrument, some investigators have divided appetitive traits into those related to food avoidance (i.e., slowness in eating, satiety responsiveness, emotional undereating, and food fussiness) and those related to food approach (i.e., enjoyment of food, food responsiveness, desire to drink, and emotional overeating) [51]. Higher scores on food approach related subscales and lower scores on food avoidant related subscales have been positively associated with weight status in children [46,54,55,56]. While generally most studies have not shown systematic differences in appetitive traits between male and female children, a few studies have reported sex differences. For example, in a cohort study of middle childhood, males from Thailand had greater enjoyment of food than females [57], however the opposite was found in a cohort of 6–7-year-old Dutch children (i.e., females higher than males) [52]. When looking more broadly across appetitive traits, male children showed greater desire to drink [57], emotional overeating [52], and food responsiveness [53]. In contrast, females showed greater avoidance behaviors (e.g., slowness of eating and satiety responsiveness) [46,58]. 

Evidence of greater food approach behaviors among male compared to female children may be in part due to differential parent feeding strategies that reinforce these behaviors. Mothers of female children report greater concern about them putting on weight [59], and therefore they may encourage greater food avoidant strategies. On the other hand, male children receive greater encouragement to eat [60,61] and are served larger portion sizes from a virtual buffet than female children [62]. These domain-specific parenting strategies [63] may encourage the development of more avid appetites among males and more food avoidant strategies among females. Therefore, it is critical for future studies to take into account the role of parents in the development of eating behaviors in males and females. 

### 2.4. Evidence of Sex-Effects on Laboratory Measures of Self-Regulatory Eating

The literature reviewed in the previous section indicates few systematic sex differences in parent-reported measures of children’s appetite. While questionnaires are convenient for capturing an overview of child behaviors, responses may be affected by the biases parents have about feeding male versus female children. Objective measures are necessary to provide additional support for the role of sex in childhood eating behaviors. In the following section, results are reviewed from studies that have used laboratory methods to characterize “self-regulatory eating”, broadly defined in this context as the ability to regulate energy consumption in response to internal or external signals. 

#### 2.4.1. Compensation Protocols

One of the most frequently used methods to assess self-regulatory eating is the compensation or preloading paradigm. Using a crossover design, children consume appetizers or “preloads” on two separate visits. Preloads are matched for taste, sensory characteristics, and often volume, but are covertly manipulated to vary in energy density (kcal per weight or volume of food or beverage) and/or macronutrient content. Participants are compelled to finish the preload and are served an ad libitum meal some time later (often 25–30 min with children) to measure consumption. Children who have “good” energy compensation can adjust their intake at the subsequent meal based on the energy content of the preload [64,65]. Poorer compensation ability has been associated with higher weight status in children [66,67,68], suggesting that performance on this measure may generalize to eating regulation more broadly. Several studies that have used this protocol in preschool children found that males have better energy compensation than females [66,67,69,70], which is consistent with some studies in adults [71,72]. Notably, other studies in preschool children do not report sex differences [73,74,75,76,77] and the individual variability in this measure is poorly understood. Of note, all the studies that have found that males compensate better than females have used beverages as a preload, raising the possibility that sex differences in energy compensation may be specific to the ability to regulate calories in liquid rather than solid form. 

The notion that sex differences around eating self-regulation are specific to beverages is further supported by studies that have tested the effect of varying the energy density of a beverage served *within* a meal. Whereas the traditional preloading study measures “satiety” by testing the extent to which a preload or snack delays hunger at the following meal, serving a beverage within a meal captures “satiation” by determining the effect of varying energy content on total meal intake. Kling and colleagues [78] tested the effect of varying the energy density (ED) of milk on satiation by conducting a crossover study where either lower—(1% fat) or higher—(3.25% fat) ED milk was served to children with a typical preschool meal served in a childcare setting. When the higher-ED milk was served, males decreased their intake of the other meal items, whereas females did not. Thus, compared to males, females were less accurate at adjusting their intake to account for additional energy consumed from the higher-ED milk. These sex differences were independent of possible confounders, including the type of milk children consumed at home, child age and body size, milk liking and preference ratings, children’s appetitive traits, and parent feeding practices. The pattern of sex differences observed in both satiety and satiation studies challenges the notion that compensatory responses are solely due to the delay between the preload and subsequent meal that allows for the release of sensory and nutrient signals that influence fullness.

#### 2.4.2. Eating in the Absence of Hunger

Eating in the absence of hunger (EAH) is a standard paradigm to assess hedonic eating [79,80,81]. It is thought to be stable through childhood [82], and is considered a phenotypic characteristic of childhood obesity [83]. Studies in preschoolers [75,83] and middle childhood [84] have found greater eating in the absence of hunger in males compared to females. However, in 5–18-year-old Hispanic children from the United States, sex differences did not persist after adjusting for energy needs [85]. Although individual differences in EAH may be partially driven by child energy needs, there is evidence that sex may moderate the relationship between EAH and outcomes such as child weight status [68,84,86,87], parental dieting characteristics, and feeding practices [80,86]. For example, some maternal behaviors such as dietary disinhibition [86] and restriction [88] are more predictive of EAH in females than in males. On the other hand, greater use of pressure to eat has been found to be a stronger predictor of EAH in males than in females [89,90]. These findings highlight the need to model the relationships between child level (i.e., sex and weight status) and parent level (i.e., feeding practices, eating styles, sex, and weight status) variables to elucidate the pathways leading to excess energy consumption in males and females. 

#### 2.4.3. Meal-Related Microstructure

Although not specifically related to the ability to regulate food intake, some investigators have referred to eating behaviors that make up meal microstructure (e.g., bite rate, eating rate, and bite size) as indicators of satiety responsiveness [91]. Of these characteristics, eating rate has been most consistently associated with weight status in adults [92] and children [93], and is therefore a target for interventions to treat obesity [94]. Observational coding of meal-time behaviors in the GUSTO cohort from Singapore showed that male children have faster eating rate (g/min), larger bite size (g/bite), and shorter oral exposure (min) than female children [95]. Similar findings have been reported in adolescents [96,97]. As masticatory development has been thought to be similar in males and females before puberty [98], it is unlikely that faster eating speed among male children can be attributed to stronger muscular force supporting the jaws. It has also been reported that eating rate has a genetic component, which may help to shed light on these differences [99]. While the research in this area is limited, the observation of sex differences in eating speed and oral processing time prior to puberty has implications for the development of personalized interventions to reduce overeating in males and females. 

In addition to the aforementioned paradigms, other measures have been considered to assess self-regulatory eating in children, for example measuring children’s intake in response to manipulations in food portion size [100,101], energy density [102], or self-serving conditions [103]. To the best of our knowledge, sex differences have not been observed in self-regulatory eating using these assessments, thus they are not discussed further in this paper. 

While there is a lack of investigations that have included sex as a primary determinant of eating behaviors, the studies reviewed are suggestive of male–female differences in food liking and intake, appetitive traits, self-regulatory eating, and meal-related microstructure. In addition, there is evidence that child weight status may moderate the relationship between sex and eating in the absence of hunger. Since these differences could impact the success of dietary and behavioral interventions [15], additional research focused on clarifying the pathways by which eating behaviors develop in males and females is needed.

## 3. Biopsychosocial Contributions to Sex Differences

This section explores possible mechanisms for the observed sex differences in children’s eating behaviors. The scope of the discussion has been limited to: (1) neural responses to food cues (a potential biological influence); (2) body image and weight concerns (potential psychological influences); and (3) parental feeding attitudes and practices (potential social influences). Additional potential influences within these biological, psychological, and social constructs are presented in Figure 1, but are not explored at length in this paper. For additional insight on mechanistic pathways in the development of childhood eating behaviors, the reader is directed to recent reviews [104,105].

In addition to serving as a framework for presenting potential mechanisms that influence the development of eating behaviors in males and females, the biopsychosocial model can help guide the planning of new studies. The model can provide insight to help in the generation of new hypotheses that can be tested to further understanding of how eating behaviors develop in males and females. In addition, it can inform the types of questionnaires and measures that should be included when planning a study and can suggest potential interactions between variables to query during statistical analyses. 

### 3.1. Neural Differences in the Response to Food Cues

One potential contribution to differences in eating behavior between male and female children is variation in neural processing of food cues. Food cues elicit responses in brain regions implicated in executive function, subjective valuation (e.g., orbitofrontal cortex), and visual processing (e.g., fusiform gyrus) [106] that are correlated with eating behaviors [107,108]. Several studies have observed sex differences in neural response to food cues. For example, in adult samples that have used functional magnetic resonance imaging (fMRI) to assess food cue reactivity, females show greater activation than males in a number of brain regions associated with executive function (i.e., dorsolateral and ventromedial prefrontal cortex) [109,110], visual processing [111] (e.g., fusiform gyrus), taste and interoceptive processing [111] (e.g., insula), and reward (e.g., caudate) [112]. To date, only one study has reported sex differences in children, although the findings contradict those from adults. Luo and colleagues [113] found that, compared to females, 7–11-year-old males had greater activation to food relative to non-food images in the right posterior hippocampus and temporal occipital fusiform cortex, regions implicated in memory and visual processing. To date, the developmental trajectory of neural response to food cues remains unclear, making it difficult to interpret the inconsistent patterns of sex differences between adult and child samples.

### 3.2. Body Image and Weight Concerns

From a young age, individual differences in eating behaviors may in part be driven by sex differences in perceived ideal and preferred body size. Sex differences in dieting and body image concerns have been consistently documented in children as young as eight years [114]; however, differences in younger children are less consistent [114,115,116]. Compared to males, school-aged females report higher levels of weight-related behaviors and concerns, including desire to lose weight [117], dieting behavior [115], level of worry about weight and thoughts about which foods might promote weight gain [115,118,119], and feelings of guilt over eating too much [118]. Females also tend to be more dissatisfied with their bodies [116,117,120,121,122] and have lower self-esteems [121,123,124]. By eight years of age, females have greater body dissatisfaction than males [114,116,117,118,122,124,125] and this tends to increase during middle childhood [124]. Overall, greater emphasis on the maintenance of an ideal body weight in females than males may encourage sex differences in eating behaviors that are adopted to achieve “the perfect figure”. 

### 3.3. Parental Feeding Styles and Practices

The greater emphasis on “thinness” as a cultural ideal in females likely encourages sex differences in parental feeding practices and attitudes directed at children. In general, parents are more concerned about weight status in female children than they are in males [63,117]; thus, they are more likely to assume an active role in training, redirecting, and encouraging desired eating behaviors in female children [63,117]. Studies have also found that male children are encouraged to eat more than female children [60,61], while females are more likely to seek parental praise and approval for meal-time behaviors [60]. In response to maternal concerns, female children are more likely than male children to change eating behaviors [125,126]. These observations could partially explain sex differences in food acceptance and intake, whereby female children show more nutritious food intake patterns than males [32]. Greater need for external attentions, such as praise, among females could mean that they are less attentive to internal signals of hunger and fullness when compared to males, which may increase their risk for disordered eating behaviors. 

The influence of controlling feeding practices, such as restriction and pressure-to-eat, have also been found to vary depending on the sex of the child. Observational coding of meals in Singapore revealed that mothers respond to faster eating in females by using more restriction and control-related prompts, but similar relationships were not found in males [127]. Greater laboratory [80,128] and parentally-reported restriction [67,129,130] have been associated with higher weight status in primarily Caucasian females, but not males. In addition, Arredondo and colleagues [131] found in Latino families that greater parental control over feeding is associated with increased reported intake of “unhealthy” foods (e.g., sodas, sugar sweetened beverages, chips, and sweetened cereals) in females, but not males. In general, mothers tend to use greater feeding control with female than male children [132]. Increased use of parental control, specifically within the domain of feeding, may weaken females’ ability to eat in response to internal satiety signals, which may ultimately increase weight gain and risk for disordered eating. Notably, these patterns have not been consistently observed across studies. Studies in both preschool children [89] and a Dutch sample in middle childhood [90] found that controlling feeding practices were associated with greater eating in the absence of hunger [89] and external and emotional eating [90] in males, but not females. Overall, the influence of child age, ethnicity, and socioeconomic status, as well as parental factors including education, weight status, and general parenting style have not been clarified and require additional investigation. 

### 3.4. Peer and Social Influences

In addition to parental influences, societal ideals related to expectations about what and how males and females should eat may also engender different eating behaviors in children. A feminine identity is characterized by eating smaller portions, consuming less meat, and preferring healthier options to maintain appearance, while a masculine eating identify is characterized by feeling full, with a focus on physical performance [133,134]. Within these ideals, female children are seen as more effective at modeling healthy behaviors than males [135,136]. Furthermore, females are also more likely to respond to modeled eating behaviors including vegetable acceptance [135] and fruit and vegetable intake [137]. The higher success of modeling and dietary interventions among females suggests a greater awareness of social expectations related to eating [138]. Moreover, greater self-control among females [139,140] may help facilitate greater uptake of these behaviors.

## 4. Applying the Biopsychosocial Model to Interpret Evidence of Sex Differences in Children’s Eating Behaviors

In the prior two sections of the paper, we reviewed evidence from the literature of sex differences in children’s eating behaviors and provided a biopsychosocial model as a framework for understanding how eating behaviors develop in males and females. A theme across the various studies reviewed is that the relationship between weight status and eating behaviors differs in males and females, and these differences may stem in part from parental feeding practices and societal pressures on ideal/acceptable body weights that differ by sex. Adding to this theme, the last section of the paper presents previously unpublished, secondary data analyses to determine the influence of sex and weight status on children’s appetitive traits and neural response to food cues. 

### 4.1. Case Study #1. Influence of Age, Sex, and Adiposity on Appetitive Traits

Although previous studies found higher food approach related behaviors among males than females [52,53,57], it is unclear how age and/or development might influence the relationship between appetitive traits and sex, as children might show higher food approach related behaviors during times rapid growth. For this reason, it is essential to understand whether the relationship between appetitive traits differs by child age and weight status. To shed light on this relationship, we examined CEBQ scores from 11 datasets collected from studies conducted at the Children’s Eating Behavior Laboratory at The Pennsylvania State University during 2012–2018. A total of 263 (M = 133; 50.6%) 3–12-year-old children had complete parent-reported anthropomorphic and CEBQ data as well as measured child anthropometrics. Males and females did not differ by age (*t*(260) = 0.553, *p* = 0.581, *d* = 0.07), body mass index (BMI)-for-age percentile (BMI%; *t*(260) = −0.859, *p* = 0.391; *d* = 0.11), race (Fisher’s *p* = 0.276), ethnicity (Fisher’s *p* = 0.999), maternal education (*t*(260) = 0.551, *p* = 0.58, *d* = 0.07;), or CEBQ subscales (*p* values ranging 0.073–0.681; see Appendix A). Although maternal education did not differ by child sex, it was used as a proxy for socioeconomic status as maternal education has been shown to be more highly associated with adiposity than income [141]. Child weight status was assessed by measuring height and weight on a digital scale (Tanita, Arlington Heights, IL, USA) and stadiometer (SECA, Chino, CA, USA) and children were categorized as either having healthy weight (BMI-for-age < 85th percentile) or overweight/obesity (BMI-for-age ≥ 85th percentile) (Table 1).

Food approach and avoidance, as measured with CEBQ, were examined separately using the same hierarchical model steps: (1) child age and maternal education; (2) a quadratic age term; (3) child sex and adiposity; (4) a sex X age interaction; and (5) a sex X adiposity interaction (Table 2). The change in model fit, R^2^, was tested at each step to determine whether the model explained significantly more variance with the added terms. Once the best model was identified, exploratory analyses examined the component subscales that contribute to the food avoidance and approach scores to determine whether the effect seen was consistent across subscales or driven by an individual subscale.

Individual differences in CEBQ avoidance and approach behaviors were best fit by different models. Child sex was not a significant predictor of avoidance for any of the models where it was included. In contrast, food approach was best modeled by including the interaction between child sex and weight status (Table 2). The interaction between sex and weight status was significant such that the association between having overweight or obesity and greater food approach was stronger for females than males. This suggests that weight status may be more predictive of food approach behaviors in females than in males. Exploratory analyses of approach subscales indicated that this finding was primarily driven by the food responsiveness subscale, which showed a suggested sex by weight status interaction (β(SE) = −0.36 (0.20), *p* = 0.073). The interactions between weight status and other CEBQ approach subscales were not significant (*p* values ranging from 0.155–0.255). Overall, these results suggest that, in female children, food responsiveness could be a better predictor of weight status than other CEBQ approach subscales, and therefore may be a target for intervention studies in this population.

### 4.2. Case Study #2. Influence of Sex on Neural Food Cue Responsivity

In a separate dataset of 7–11-year-old children who had participated in a study on the neural determinants of food portion size and energy density [108,142], we followed up findings from Luo and colleagues [113] to investigate potential sex differences in children’s food cue reactivity. As with the first case study, child weight status was treated as a key moderating factor. Males (*N* = 22) and females (*N* = 25) did not differ by age (*t*(45) = 0.89, *p* = 0.378, *d* = 0.260), BMI-for-age percentile (*t*(45) = 0.125, *p* = 0.901, *d* = 0.036), race (Fisher’s *p* = 0.095), ethnicity (Fisher’s *p* = 0.456), or maternal years of education (*t*(45) = −0.045, *p* = 0.964, d = 0.013) (Table 1).

On the day of the MRI, children arrived after a 2-h fast and were scanned during a usual meal-time. Before and after the scan, children rated fullness level on a validated, pictorial visual analog scale [143]. Children were imaged at 3T (MAGNETOM Trio) with a T1-weighted structural (MPRAGE) sequence and a T2*-sensitive gradient echo pulse sequence (see Appendix A for image acquisition parameters). Food images were presented using MATLAB Version 8 [144] and viewed through a mirror mounted on the head coil using a magnet-compatible projector. The protocol for task design and image development has been reported elsewhere [108,142]. In brief, children viewed a total of 180 images (120 food, 30 furniture, and 30 scrambled images) presented in block design. The food cues differed in portion size (large or small) and energy density (high-ED or low-ED). High-ED foods were >1.5 kcal/gram and included French fries, chicken nuggets, cookies, and pizza. Low-ED foods were <1.5 kcal/gram and included grilled chicken, carrots, broccoli, and apples. Data were preprocessed and analyzed using Analysis of Functional NeuroImages (AFNI) [145] using standard preprocessing steps (see Appendix A for details). Four participants (3 male and 1 female) were excluded due to excessive motion (defined as fewer than 4/6 usable runs; see Appendix A for motion and outlier criteria). For each subject, a general linear model was constructed including 6 parameters of interest (i.e., one for each image condition) and 12 parameters of no-interest to control for motion (see Appendix A). Group analyses were then conducted using energy density contrasts (high-ED – low-ED) derived from parameter estimates for each portion size condition separately, as well as a composite (i.e., across both portion sizes). Multiple comparisons were controlled by using Monte-Carlo simulations ([146] *p* < 0.001; *k* = 29) using AFNI’s 3dClustSim to achieve a final *p* < 0.05. 

As there was no main effect of portion size, or a portion size × sex interaction on neural response to high- or low-ED cues (see Appendix A), the remaining group analyses focused on the ED contrast collapsed across portion size. An analysis of covariance (ANCOVA; 3dMVM [147]) showed a significant sex × BMI z-score interaction in right superior temporal gyrus, extending to both parahippocampal and fusiform gyri F(1,39) = 29.21; peak: x = −37.5, y = 37.5, z = 7.5; k-173; Figure 2A). Post-hoc correlations confirmed a significant positive association between BMI z-score and neural response to higher than lower ED food images in females (*r* = 0.598; *p* = 0.002), while in males this relationship was negative (*r* = −0.667; *p* = 0.002) (Figure 2B). There was no evidence for a main effect of BMI z-score or sex. Although pre- and post-scan fullness differed in males and females, the same pattern of results was seen when controlling for fullness ratings and when analyses included ED contrasts for each portion size (see Appendix A) (Figure 2A,B).

Although preliminary, these results suggest that increased weight status in female children is positively related to neural engagement to high- relative to low-ED food cues in regions typically associated with contextual processing (i.e., parahippocampal gyrus) and visual object recognition (i.e., fusiform gyrus), while in male children the opposite pattern was observed. While additional studies are needed to confirm these findings, they suggest that weight status in female children may be more associated with differential patterns of food-cue related brain activation than weight status in male children. Questions for additional investigation include determining whether brain alterations precede or follow the development of excess weight and understanding the behavioral implications for these neural responses.

## 5. Summary and Conclusions

In this paper, we review evidence of sex differences in children’s eating behaviors and present new data showing that sex and weight status interact to differentially influence appetitive traits and neural response to food images in males and females. In the reviewed literature, we identified sex differences in food acceptance, food intake, appetitive traits, and laboratory measures of self-regulatory eating. In addition, new analyses showed that child weight status interacts with sex to influence appetitive traits such that food approach behaviors (i.e., food responsiveness) are stronger predictors of increased weight status in females than in males. Similarly, in a separate cohort of 7–11-year-olds, we found that sex and weight status interact to influence children’s neural responses to food images that vary in energy density. In females, greater activation to higher energy food cues in brain regions implicated in contextual processing, memory, and object recognition was positively related to weight status, while the opposite pattern was observed in males. Although we cannot fully discount the possibility that some of the observed differences are driven by physiological changes that occur with puberty, the focus on children under 11 years of age likely reduces these influences. The evidence presented underscores the need to study the etiology and implications of sex differences in children’s eating behaviors. 

Despite inconsistencies across the literature, a few consistent themes are apparent. First, sex differences in children’s eating behaviors were more often found in school-aged children. Few consistent differences in eating behaviors were identified among infants and toddlers. It is possible that differences are present in younger children but are unable to be measured due to methodological limitations. Perhaps more likely, however, is that these patterns arise during childhood due to differential parenting practices and social influences directed at males and females. Second, female children tend to report liking and eating more foods that are lower in energy density and higher in critical nutrients (i.e., fruits and vegetables) than males. Due to the lack of clear biological differences in taste anatomy [148], these differences are also likely to be influenced by parent, peer, and societal factors. Importantly, the self-report nature of most of this literature highlights the need to confirm these findings with more objective measures of eating behavior. Third, sex differences in appetitive traits, EAH, and parental feeding attitudes are influenced by complex interactions with child weight status. In general, parents are more concerned about excess weight in female compared to male children. As a result, they likely feed children differently depending not only on the sex of the child, but also their perception that the child is at risk for developing overweight. It is likely that parental characteristics, such as dieting history, cognitive restraint, socioeconomic status, and weight status, influence the relationship between child sex and eating behaviors, highlighting the need to conduct larger studies that are sufficiently powered to query three-way interactions (e.g., child sex × child weight status × parent weight history).

## 6. Recommendations for Future Research

When planning and reporting on future studies, it is important that researchers clearly define the constructs of sex and gender, in terms of how they are measured and reported. In addition, in studies that statistically control for sex as a covariate, it would be helpful for researchers to report applicable estimates, coefficients, and *p*-values for covariates, either in the manuscript or in Appendix A. This would facilitate the ability to conduct systematic reviews on this topic. Moreover, research in children, especially infants and preschool children, should utilize objective and observational measures of children’s eating behaviors and intake when possible to limit the influence of parental beliefs and perceptions along with probable response bias for questionnaires. Lastly, in regards to intervention efforts for obesity, sex or gender should be considered when determining target behaviors as well as evaluating the impact of the intervention on primary and secondary outcomes. Together, these recommendations will help advance our understanding of the role that sex and gender play in the development of weight and eating disorders. 

Caution is recommended when interpreting the findings discussed, both from the literature and the new analyses presented. First, the majority of studies that have reported sex differences were not designed to detect sex as a primary determinant of outcomes; thus, it is not possible to rule out chance findings. Second, among the studies that did not report differences, sex was often controlled for as a covariate, but results for main outcomes were not stratified and reported by sex. This makes it difficult to determine whether primary eating behavior outcomes differed in males and females and limits the ability to conduct meta-analyses across studies. Third, determining the underlying mechanisms for sex differences in eating behavior is complicated by the lack of clarity in how sex and gender are defined in the literature. A concern moving forward is that researchers will overgeneralize findings by developing separate intervention approaches for males and females without considering that sex and gender are non-binary, multidimensional constructs. To avoid this type of overgeneralization, we caution against using sex or gender as the basis to group participants prior to assigning treatments. Instead, sex and gender should be measured and considered such as other individual subject characteristics and used to provide information to help phenotype risk groups.

## Figures and Tables

**Figure 1 nutrients-11-00682-f001:**
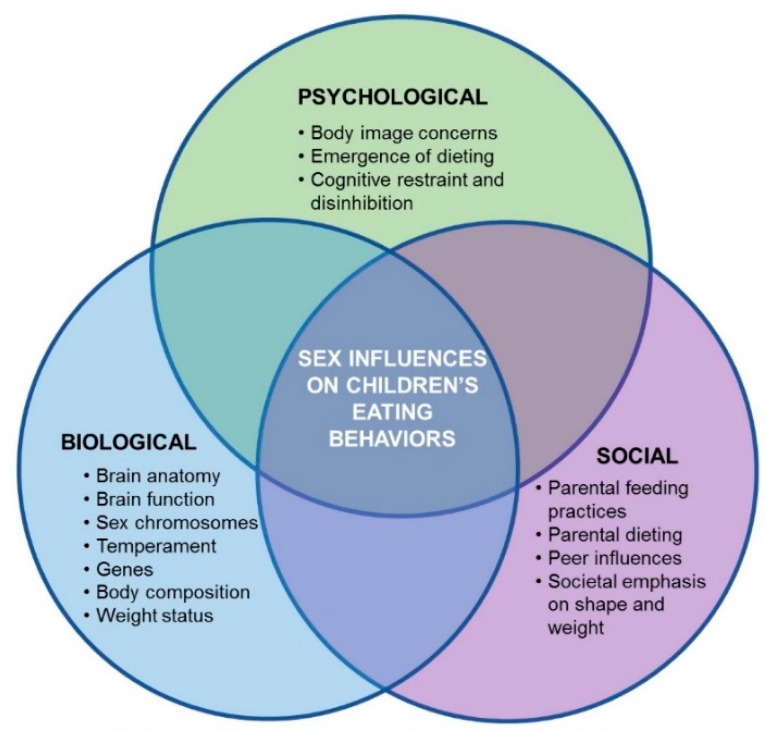
Biopsychosocial model of sex effects on children’s eating behaviors. Potential biological influences could come from differences in brain anatomy or brain function that arise early in development, effects due to sex chromosomes, temperament, genes, or differences in body composition and/or weight status that can influence food intake regulation. Psychological influences include body image concerns, dieting, and cognitive restraint and disinhibition, typically observed more frequently in females than males. Social influences include differences in parental feeding practices directed at males and females, parental dieting, peer influences, and societal emphasis on “thinness” in females and “bigness” in males.

**Figure 2 nutrients-11-00682-f002:**
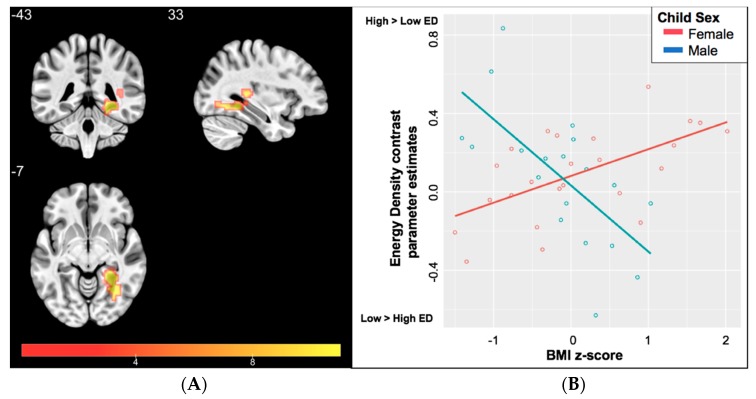
(**A**) Statistical parametric map (F-statistic) of the interaction between BMIz and child sex on neural responses to high-ED compared to Low-ED food cues. Cluster extends from the right superior temporal gyrus into the parahippocampal and fusiform gyri. (**B**) Extracted energy density contrast (high-ED–low-ED) parameter estimates, illustrating increased activation to high-ED compared to low-ED food cues for girls with BMIz greater than the 50th percentile and increased activation to high-ED compared to low-ED food cues for boys with BMIz greater below the 50th percentile.

**Table 1 nutrients-11-00682-t001:** Demographic Characteristics of children enrolled in studies that assessed sex differences in appetitive traits^a^ and neural responses to food cues^b^.

	CEBQ ^a^		Fmri ^b^	
	Males (*n* = 133)	Females (*n* = 130)	Males (*n* = 20)	Females (*n* = 25)
Age (years)	7.40 (2.28)	7.56 (2.10)	8.75(0.99)	9.06(1.34)
BMI percentile	61.53 (29.06)	58.50 (28.20)	52.50(27.12)	53.57(30.93)
Maternal Ed. (years)	16.19 (2.63)	16.35 (2.71)	16.91(2.49)	16.88(1.90)
Weight Status (*n*)				
Obese/Overweight	43	27	3	6
Healthy Weight	90	103	19	19
Ethnicity (*n*)				
Not Hispanic/Latinx	94	84	20	25
Hispanic/Latinx	4	4	1	0
Not Reported	35	35	1	0
Race (*n*)				
Black/African American	6	2	2	0
White	119	112	19	25
Other	7	4	1	0
Not Reported	1	2	0	0
SES (*n*)				
>$100,000	16	19	7	5
$51,000–$100,000	30	29	11	15
≤$50,000	18	18	3	5
Not Reported	69	64	1	0

Means (SD) reported for Age, BMI percentile, and Maternal Education. Weight Status categories defined by BMI percentile: Obese/Overweight ≥ 85th percentile; Healthy Weight < 85th percentile. BMI, body-mass index; CEBQ, Child Eating Behaviors Questionnaire Sample; fMRI, functional Magnetic Resonance Imaging Sample. ^a^ Sample assessing appetitive traits in case study #1; ^b^ Sample assessing neural responses to food cues in Case Study #2.

**Table 2 nutrients-11-00682-t002:** Hierarchical Regression for Approach and Avoidance Scales of the Child Eating Behavior Questionnaire.

Food Approach															
	1			2			3			4			5		
	B	SE	β	B	SE	β	B	SE	β	B	SE	β	B	SE	β
Maternal Education	−0.005	0.01	−0.027	−0.005	0.01	−0.028	−0.005	0.01	−0.025	−0.005	0.01	−0.025	−0.003	0.01	−0.015
Age	0.008	0.01	0.034	−0.003	0.09	−0.015	0.0003	0.01	−0.001	0.001	0.02	0.003	−0.002	0.01	−0.008
Age-squared				0.001	0.01	0.050	--	--	--	--	--	--	--	--	--
Weight Status							0.288	0.07	0.572 ***	0.288	0.07	0.288 ***	0.455	0.11	0.902 ***
Sex							−0.119	0.06	−0.002	−0.107	0.22	−0.237	−0.046	0.07	−0.091
Sex X Age										−0.002	0.03	−0.007	--	--	--
Sex X Weight Status													−0.286	0.14	−0.567 *
R^2^			0.002			0.002			0.071			0.071			0.086
∆ R^2^ F						^1^ 0.017			^1^ 9.59 ***			^3^ 0.003			^3^ 4.21 *
Food Avoidance															
Maternal Education	0.017	0.01	0.094	0.016	0.01	0.093	0.017	0.01	0.094	0.017	0.01	0.097	0.017	0.01	0.093
Age	−0.029	0.01	−0.137 *	−0.072	0.08	−0.339	−0.027	0.01	−0.126 *	−0.035	0.02	−0.165	−0.026	0.01	−0.125 *
Age-squared				0.003	0.01	0.205	-	-	-	-	-	-	-	-	-
Weight Status							−0.073	0.07	−0.156	−0.071	0.07	0.151	−0.087	0.10	−0.184
Sex							0.068	0.06	0.143	−0.049	0.21	0.143	-	-	-
Sex X Age										0.015	0.03	0.072	0.062	0.07	0.131
Sex X Weight Status													0.023	0.13	0.049
R^2^			0.029			0.030			0.038			0.039			0.038
∆ R^2^ F						^1^ 0.303			^1^ 1.137			^1^ 0.0.87			^1^ 0.765

Standardized coefficients (B) and standard errors are presented along with the unstandardized coefficients (β) for model Steps 1–5. For change in R-square, the number in the brackets indicate which model step it was tested against. * *p* < 0.05, *** *p* < 0.001. ^1^ Model was tested against model 1; ^3^ Model was tested against model 3

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
