# Peer review of "A Biopsychosocial Model of Sex Differences in Children’s Eating Behaviors"

_nutrients, 2019, doi:10.3390/nu11030682_

Round 1

Reviewer 1 Report

General Comments:

In “A biopsychosocial model of sex differences in children’s eating behaviours”, Keller and colleagues present evidence for sex differences in aspects of children’s eating behaviour (primarily focusing on 0-11 years old), followed by a summary of key biological and psycho-social factors that might explain some of these differences.  The authors clearly distinguish between the constructs of sex and gender and make a clear case for why a better understanding of how sex and gender impact the development of eating behaviours during childhood is important. Sex differences in children’s eating behaviours have not really been systematically studied. In this regard, the current paper provides a starting point and framework within which this line of enquiry should be taken forward.  However, some refinement to the structure and clarity around the evidence selected is necessary to improve its impact.

The authors acknowledge that the majority of the sex differences reported in the literature were not set out to be tested. Inevitably, there is a general sense of inconsistency within many of the findings, and it is not always clear whether meaningful conclusions can be drawn from the different sections. However, part of the inconsistency also occurs because it is not clear why some aspects of eating behaviour are discussed, and others are not (this is most true for sections 1.1-1.6). For instance, there is a section on “food acceptance/preference” but no mention of neophobia or picky eating, and there is a section on self-regulatory eating behaviours that is restricted to compensation trails primarily. There is also a section on the CEBQ and EAH, but it is unclear why these are combined. This is not to say that the review needs to cover all possible aspects of children’s eating behaviours, but rather that the reason for focusing on the different behaviours should be justified and cohesive. It feels like the eating behaviours selected are those that had the most sex differences, but this would mean that the narrative of the paper is really being driven by what could be (in some cases) chance findings, which would only make sense if there are some differences that are very consistent.  To me, it would more impactful to focus on two or three aspects of eating behaviours that have been identified as important determinants of children’s energy intake and/or growth outcomes (or alternatively select behaviours that are generally considered to show sex differences in adulthood), and then systematically consider whether sex differences have/have not been reported in children, and what biological, social and psychological factors could be driving them. Aspects of eating behaviour that do appear to be relatively consistent across sexes should be leveraged more to help place some of the reported differences in perspective and draw clearer conclusions (see related point about self-regulation in the specific comments below).   

Finally, the authors’ also present new data to shed light on sex differences in parent reported appetitive traits (CEBQ) and neural responses to food cues. These data are interesting but I found that the necessary description of the methods, analyses and results distracted from the flow of reading quite a bit. Is it possible to present these separately? Perhaps as ‘case studies’ where the outcomes could be interpreted in the context of the proposed biopsychosocial model?

Specific comments:

Sections 1.1-1.3 – There should be more justification as to why other related aspects of perception and behaviours are not considered. What about differences in odour perception or food neophobia? Both could be argued as having an evolutionary relevance to the development of eating behaviours, so it would be interesting to comment on whether there are/are not sex based difference noted in the literature. 

Section 1.3 – Consider changing ‘food intake’ to ‘dietary patterns’ in the title, or widen the scope of the research discussed. To me, food intake also relates to meal size and factors affecting total energy intake (e.g. portion size provided), rather than just the types of foods children eat more/less off.

119-123 – The conclusion reached here is good but seems to come a bit early. It would make more sense at the end of section 1.3 (or at least come back to it here), where a broader conclusion about the link (or lack of) between taste anatomy and sex differences in preferences and dietary patterns would round off these three interrelated sections.

Section 1.4 – Sex effects in self-regulatory eating – This section seems quite selective. The authors restrict the self-regulatory eating to compensation ability measured in preload studies, and one of the author’s satiation studies in preschool children. Responsivity to variations in portion size, dishware size and energy density where intake of the manipulated food is the outcome measure also assess aspects of self-regulatory eating. Are there systematic sex-differences in these eating behaviours?  If there are/are not, this is worth mentioning.  I’m also not convinced by the idea that sex-dependent differences in compensation abilities may be specific to energy consumed in liquid form (Lines 196-201). What is the reasoning behind this? The possible explanations given on lines 218-219 are very vague. We know that generally adults are worse at compensating for the energy in liquid form in preload studies, compared to solid and semisolid foods (see Almiron-Roig et al, Nutrition Reviews, 2013), but there was no significant interaction with participant sex, indicating that (in adults at least) this relationship appears to be similar for males/females. What about some of the studies that have tested for and reported no sex differences in other aspects of eating behaviour linked to self-regulation (e.g. Savage  et al, 2012, AJCN , Do children eat less at meals when allowed to serve themselves?; Fisher et al 2007, Obesity, Effects of Age on Children's Intake of Large and Self-selected Food Portions, just as examples).

Section 1.5 – It is unclear why appetitive traits (CEBQ) and EAH have been presented together in this section. The authors indicate that the EAH research provides laboratory-based evidence for over consumption (line 278-279), but so would a number of other behaviours (portion size literature, responses to energy density within-meal… etc).

Section 1.6 – Microstructure. It might also be worth considering the review by Le Reverend and others (2014, British Journal of Nutrition, Anatomical, functional, physiological and behavioural aspects of the development of mastication in early childhood), who comment that masticatory development does not appear to differ for male and female children until after puberty.

Line 302-307 – This appears to be a conclusion for the sections 1.1-1.6.  This would be better as a standalone section, or as an introduction to section 2.

Section 2.3 – Parental influences – There is a recent paper from the GUSTO cohort that comments on sex differences in oral processing behaviours and their link to parental feeding practices (Fogel, McCrickerd et al, 2018, Maternal and Child Nutrition, Prospective associations between parental feeding practices and children's oral processing behaviours). Though again this study was not set up or powered to make a robust assessment of sex-based interactions.  

Minor points:

Lines 45-60 – Are there differences in growth trajectories between females and males from infancy to later childhood? This would be interesting to comment on here.

Line 51 – The statement that adult males carry fat around their abdomen while females are metabolically protected should be moderated. Although there is an overall tendency for the adiposity distribution described, within women there are large differences in adiposity distribution based on ethnicity which should not be dismissed. Females from south Asia, for instance, are not necessarily metabolically protected by their body composition.      

Title – Section 1 – Would “Evidence for sex differences in children’s eating behaviour” read better?

Line 103 – lower thresholds – for what? Taste preference or detection, or both?

Line 292 – Number (1.6) missing from section

Author Response

In “A biopsychosocial model of sex differences in children’s eating behaviours”, Keller and colleagues present evidence for sex differences in aspects of children’s eating behaviour (primarily focusing on 0-11 years old), followed by a summary of key biological and psycho-social factors that might explain some of these differences.  The authors clearly distinguish between the constructs of sex and gender and make a clear case for why a better understanding of how sex and gender impact the development of eating behaviours during childhood is important. Sex differences in children’s eating behaviours have not really been systematically studied. In this regard, the current paper provides a starting point and framework within which this line of enquiry should be taken forward.  However, some refinement to the structure and clarity around the evidence selected is necessary to improve its impact.

 Response: We very much appreciate your helpful feedback to further refine our paper. Our goal was to serve as a starting point to initiate this discussion and further research in this field. The reorganization and refinement of the paper (as suggested below) has helped clarify our message.

The authors acknowledge that the majority of the sex differences reported in the literature were not set out to be tested. Inevitably, there is a general sense of inconsistency within many of the findings, and it is not always clear whether meaningful conclusions can be drawn from the different sections. However, part of the inconsistency also occurs because it is not clear why some aspects of eating behaviour are discussed, and others are not (this is most true for sections 1.1-1.6). For instance, there is a section on “food acceptance/preference” but no mention of neophobia or picky eating, and there is a section on self-regulatory eating behaviours that is restricted to compensation trails primarily. There is also a section on the CEBQ and EAH, but it is unclear why these are combined. This is not to say that the review needs to cover all possible aspects of children’s eating behaviours, but rather that the reason for focusing on the different behaviours should be justified and cohesive. It feels like the eating behaviours selected are those that had the most sex differences, but this would mean that the narrative of the paper is really being driven by what could be (in some cases) chance findings, which would only make sense if there are some differences that are very consistent.  To me, it would more impactful to focus on two or three aspects of eating behaviours that have been identified as important determinants of children’s energy intake and/or growth outcomes (or alternatively select behaviours that are generally considered to show sex differences in adulthood), and then systematically consider whether sex differences have/have not been reported in children, and what biological, social and psychological factors could be driving them. Aspects of eating behaviour that do appear to be relatively consistent across sexes should be leveraged more to help place some of the reported differences in perspective and draw clearer conclusions (see related point about self-regulation in the specific comments below).   

 Response: We agree that this is an important concern. In the revised paper, we have made a clearer distinction on how and why we have selected the eating behaviors that we did (see Line 100-106). We also deleted some of the sections that were in the previous version (e.g., oral sensory responses) and added in sections on pickiness/neophobia. We reorganized the paper substantially so that now survey-based measures of appetite (i.e., pickiness and appetitive traits) are separated from laboratory based measures of self-regulatory eating (i.e., compensation, eating speed, and eating in the absence of hunger). These substantial changes improved the clarity of the manuscript.

Finally, the authors’ also present new data to shed light on sex differences in parent reported appetitive traits (CEBQ) and neural responses to food cues. These data are interesting but I found that the necessary description of the methods, analyses and results distracted from the flow of reading quite a bit. Is it possible to present these separately? Perhaps as ‘case studies’ where the outcomes could be interpreted in the context of the proposed biopsychosocial model?

Response: This was a very good idea. We followed the reviewer’s suggestion and in the revised version, the new data is presented in a separate section of the paper, interpreted from the context of the biopsychosocial model.

Specific comments:

Sections 1.1-1.3 – There should be more justification as to why other related aspects of perception and behaviours are not considered. What about differences in odour perception or food neophobia? Both could be argued as having an evolutionary relevance to the development of eating behaviours, so it would be interesting to comment on whether there are/are not sex based difference noted in the literature. 

 Response: Thanks for this suggestion. We had originally included a section on neophobia, but ended up cutting this from the final draft. However, the point is well-taken that we were not clear in how we defined “eating behavior” and why we selected the specific behaviors we did. We have tried to provide a more transparent definition of this in the revised manuscript (line 100-106). We have also integrated back the section on food neophobia/picky eating.

Section 1.3 – Consider changing ‘food intake’ to ‘dietary patterns’ in the title, or widen the scope of the research discussed. To me, food intake also relates to meal size and factors affecting total energy intake (e.g. portion size provided), rather than just the types of foods children eat more/less off.

 Response: This is a good point. We have changed this section to “dietary patterns.”

119-123 – The conclusion reached here is good but seems to come a bit early. It would make more sense at the end of section 1.3 (or at least come back to it here), where a broader conclusion about the link (or lack of) between taste anatomy and sex differences in preferences and dietary patterns would round off these three interrelated sections.

 Response: This section has been removed from the revision.

Section 1.4 – Sex effects in self-regulatory eating – This section seems quite selective. The authors restrict the self-regulatory eating to compensation ability measured in preload studies, and one of the author’s satiation studies in preschool children. Responsivity to variations in portion size, dishware size and energy density where intake of the manipulated food is the outcome measure also assess aspects of self-regulatory eating. Are there systematic sex-differences in these eating behaviours?  If there are/are not, this is worth mentioning.  I’m also not convinced by the idea that sex-dependent differences in compensation abilities may be specific to energy consumed in liquid form (Lines 196-201). What is the reasoning behind this? The possible explanations given on lines 218-219 are very vague. We know that generally adults are worse at compensating for the energy in liquid form in preload studies, compared to solid and semisolid foods (see Almiron-Roig et al, Nutrition Reviews, 2013), but there was no significant interaction with participant sex, indicating that (in adults at least) this relationship appears to be similar for males/females. What about some of the studies that have tested for and reported no sex differences in other aspects of eating behaviour linked to self-regulation (e.g. Savage  et al, 2012, AJCN , Do children eat less at meals when allowed to serve themselves?; Fisher et al 2007, Obesity, Effects of Age on Children's Intake of Large and Self-selected Food Portions, just as examples).

 Response: We have made several changes to this section that we feel have appropriately responded to the reviewers concerns. 1) We have changed the title of this section to “Evidence of sex effects on laboratory measures of self-regulatory eating” and included compensation protocols, eating in the absence of hunger, and meal-related microstructure. 2) We acknowledged that there are other measures of self-regulatory eating that have not shown sex-differences (i.e., portion size response, self-serving, etc). 3) We have limited the speculations about sex-differences in compensatory responses being specific to beverages.

Section 1.5 – It is unclear why appetitive traits (CEBQ) and EAH have been presented together in this section. The authors indicate that the EAH research provides laboratory-based evidence for over consumption (line 278-279), but so would a number of other behaviours (portion size literature, responses to energy density within-meal… etc).

Response: We have reorganized the paper so that EAH is included with other measures of self-regulatory eating assessed in the laboratory

Section 1.6 – Microstructure. It might also be worth considering the review by Le Reverend and others (2014, British Journal of Nutrition, Anatomical, functional, physiological and behavioural aspects of the development of mastication in early childhood), who comment that masticatory development does not appear to differ for male and female children until after puberty.

Response: We have included reference to the paper by Le Reverend. Thanks for this suggestion.

Line 302-307 – This appears to be a conclusion for the sections 1.1-1.6.  This would be better as a standalone section, or as an introduction to section 2.

Response: This section has been removed in the revision.

Section 2.3 – Parental influences – There is a recent paper from the GUSTO cohort that comments on sex differences in oral processing behaviours and their link to parental feeding practices (Fogel, McCrickerd et al, 2018, Maternal and Child Nutrition, Prospective associations between parental feeding practices and children's oral processing behaviours). Though again this study was not set up or powered to make a robust assessment of sex-based interactions.  

 Response: Thanks for this suggestion. We had this reference in an earlier version, but it was unfortunately omitted during rewrites. We have included it in the resubmission.

Minor points:

Lines 45-60 – Are there differences in growth trajectories between females and males from infancy to later childhood? This would be interesting to comment on here.

 Response: We commented briefly on differences in body composition from birth.

Line 51 – The statement that adult males carry fat around their abdomen while females are metabolically protected should be moderated. Although there is an overall tendency for the adiposity distribution described, within women there are large differences in adiposity distribution based on ethnicity which should not be dismissed. Females from south Asia, for instance, are not necessarily metabolically protected by their body composition.      

 Response: This is a good point. We have revised accordingly by softening this statement.

Title – Section 1 – Would “Evidence for sex differences in children’s eating behaviour” read better?

 Response: Thanks for this change. We have revised.

Line 103 – lower thresholds – for what? Taste preference or detection, or both?

 Response: This section has been deleted.

Line 292 – Number (1.6) missing from section

Response: This has been corrected.

Submission Date

17 January 2019

Date of this review

12 Feb 2019 03:14:41

Reviewer 2 Report

The paper addresses the issues of eating behavior in children, including whether there are apparent sex differences in the behavior. In obesity development, eating behavior is of high interest. As there are known to be more women than men that are obese, characterizing eating behavior in early childhood could provide valuable tools in obesity treatment, and hopefully in a more target prevention.

Minor:

Please start the chapter numbering in the Introduction-section.

Table 1: The table text should explain more thoroughly the content of the table, as you should be able to understand the table without reading the paper. Please change this.

 Figure 1: Please use the same font in the figure text as in the rest of the paper.

Major:

It is not evident whether the data presented in supplementary materials, and later presented in Table 1 is unpublished data and from which source/researcher?

 Any description on how and when the literature search was performed is lacking. Transparency in research is important, and lack of a method section is a major issue. You should follow the PRISMA guidelines for a structured review, and they clearly state that you should provide a thorough method section. Could you please provide this?

Suggestion: The conclusion and further research section could be two different chapters. That would make it easier for the reader. The conclusion part should be very short.

Author Response

Reviewer 2

The paper addresses the issues of eating behavior in children, including whether there are apparent sex differences in the behavior. In obesity development, eating behavior is of high interest. As there are known to be more women than men that are obese, characterizing eating behavior in early childhood could provide valuable tools in obesity treatment, and hopefully in a more target prevention.

Minor:

Please start the chapter numbering in the Introduction-section.

Response: This change has been made.

Table 1: The table text should explain more thoroughly the content of the table, as you should be able to understand the table without reading the paper. Please change this.

Response: We have included a more substantial title for Table 1.

 Figure 1: Please use the same font in the figure text as in the rest of the paper.

 Response: We are unsure about what font will be used in the Nutrients print, so we will wait on the proofs to make this change.

Major:

It is not evident whether the data presented in supplementary materials, and later presented in Table 1 is unpublished data and from which source/researcher?

Response: We have clarified in the revision that these are new secondary data analyses that have not previously been published.

 Any description on how and when the literature search was performed is lacking. Transparency in research is important, and lack of a method section is a major issue. You should follow the PRISMA guidelines for a structured review, and they clearly state that you should provide a thorough method section. Could you please provide this?

Response: The purpose of this paper was to provide a narrative review and present new data. We intentionally did not conduct a systematic review as we don’t believe the literature is in a place to do so. We have clarified the purpose of the review (line 98-106) in the resubmission. As this was not done as a systematic review, we cannot provide a methods section in Prisma format.

Suggestion: The conclusion and further research section could be two different chapters. That would make it easier for the reader. The conclusion part should be very short.

Response: This was a good suggestion. We have split these sections.

Submission Date

17 January 2019

Date of this review

24 Jan 2019 12:37:51

Round 2

Reviewer 1 Report

Thank you to the authors for their responses. My comments have been addressed. The selection of the eating behaviours reported is now clearer and justified. The reorganisation of the structure and addition/removal of certain sections has improved the papers readability and clarity. I agree that the paper provides a good starting point for future research, and highlights important issues to consider in the study of sex differences in children’s eating behaviour and the development of strategies for obesity prevention/management.

I have two minor points:

Lines 374-376 – Sentence beginning “Increased use of parental control…”. Given that this point is specific to females, would it not make more sense to link parental control to disordered eating rather than obesity risk? The authors point out that eating disorders are more prevalent in females, while obesity is more prevalent in males (reported lines 61-64)…

Typo - Lines 386-387 – “identify” should be identity ?

Author Response

Thank you to the authors for their responses. My comments have been addressed. The selection of the eating behaviours reported is now clearer and justified. The reorganisation of the structure and addition/removal of certain sections has improved the papers readability and clarity. I agree that the paper provides a good starting point for future research, and highlights important issues to consider in the study of sex differences in children’s eating behaviour and the development of strategies for obesity prevention/management. 

Response: We thank the reviewer for their feedback and comments.

I have two minor points:

Lines 374-376 – Sentence beginning “Increased use of parental control…”. Given that this point is specific to females, would it not make more sense to link parental control to disordered eating rather than obesity risk? The authors point out that eating disorders are more prevalent in females, while obesity is more prevalent in males (reported lines 61-64)… 

Response: Thanks for this point. This has been fixed.

Typo - Lines 386-387 – “identify” should be identity ?

Response: Thanks, this has been fixed.

Reviewer 2 Report

Please check if your tables look correct, there are some minor "flaws" in Table 2. 

Author Response

Please check if your tables look correct, there are some minor "flaws" in Table 2. 

Response: Table 2 errors have been fixed.